# Pharmacometrics to Evaluate Dosing of the Patient-Friendly Ivermectin CHILD-IVITAB in Children ≥ 15 kg and <15 kg

**DOI:** 10.3390/pharmaceutics16091186

**Published:** 2024-09-07

**Authors:** Klervi Golhen, Michael Buettcher, Jörg Huwyler, John van den Anker, Verena Gotta, Kim Dao, Laura E. Rothuizen, Kevin Kobylinski, Marc Pfister

**Affiliations:** 1Pediatric Pharmacology and Pharmacometrics, University Children’s Hospital Basel (UKBB), University of Basel, 4056 Basel, Switzerland; michael.buettcher@luks.ch (M.B.); jvandena@childrensnational.org (J.v.d.A.); verena.gotta@ukbb.ch (V.G.); marc.pfister@ukbb.ch (M.P.); 2Pediatric Infectious Diseases, Children’s Hospital of Central Switzerland (KidZ), Lucerne Cantonal Hospital, 6000 Luzern, Switzerland; 3Faculty of Health Sciences and Medicine, University of Lucerne, 6000 Luzern, Switzerland; 4Department of Pharmaceutical Sciences, Division of Pharmaceutical Technology, University of Basel, 4056 Basel, Switzerland; joerg.huwyler@unibas.ch; 5Division of Clinical Pharmacology, Children’s National Hospital, Washington, DC 20001, USA; 6Clinical Pharmacology Service, Lausanne University Hospital, University of Lausanne, 1011 Lausanne, Switzerland; kim.dao@chuv.ch (K.D.); laura.rothuizen@chuv.ch (L.E.R.); 7Mahidol Oxford Tropical Medicine Research Unit (MORU), Bangkok 10400, Thailand; kevin@tropmedres.ac

**Keywords:** ivermectin, STROMECTOL^®^, dosing, pharmacometrics, absorption, variability, orodispersible tablet (ODT), TIP-based technology, oral drug delivery, novel delivery systems

## Abstract

The antiparasitic drug ivermectin is approved for persons > 15 kg in the US and EU. A pharmacometric (PMX) population model with clinical PK data was developed (i) to characterize the effect of the patient-friendly ivermectin formulation CHILD-IVITAB on the absorption process and (ii) to evaluate dosing for studies in children < 15 kg. Simulations were performed to identify dosing with CHILD-IVITAB associated with similar exposure coverage in children ≥ 15 kg and < 15 kg as observed in adults receiving the reference formulation STROMECTOL^®^. A total of 448 ivermectin concentrations were available from 16 healthy adults. The absorption rate constant was 2.41 h^−1^ (CV 19%) for CHILD-IVITAB vs. 1.56 h^−1^ (CV 43%) for STROMECTOL^®^. Simulations indicated that 250 µg/kg of CHILD-IVITAB is associated with exposure coverage in children < 15 kg consistent with that observed in children ≥ 15 kg and adults receiving 200 µg/kg of STROMECTOL^®^. Performed analysis confirmed that CHILD-IVITAB is associated with faster and more controlled absorption than STROMECTOL^®^. Simulations indicate that 250 µg/kg of CHILD-IVITAB achieves equivalent ivermectin exposure coverage in children < 15 kg as seen in children ≥ 15 kg and adults.

## 1. Introduction

Ivermectin has been a cornerstone in the treatment and control of parasitic infections since its introduction in the 1980s, providing broad-spectrum efficacy against a wide range of conditions including lymphatic filariasis, onchocerciasis, head lice, intestinal helminths, strongyloidiasis, and scabies [1]. Ivermectin is on the World Health Organization’s List of Essential Medicines, underscoring its significance in global health [2]. Approximately 400 million ivermectin treatments are distributed annually by mass drug administration (MDA) to control and eliminate onchocerciasis and lymphatic filariasis [3]. In addition, since 2023, a conventional ivermectin tablet formulation (Subvectin) has been registered in Switzerland for the treatment of scabies in adults, and high priority has been given to research on child-friendly treatment modalities [4,5]. Despite its extensive use in adults, there is a significant gap in knowledge regarding its pharmacokinetic profile in young children, particularly those with a weight of less than 15 kg. Children weighing less than 15 kg are excluded from official MDA treatment programs; thus, they do not receive the benefits of ivermectin to control numerous neglected tropical diseases (NTDs) that afflict young children [6,7,8]. Further, this contraindication leads to off-label use of ivermectin without a robust evidence base for appropriate dosing in children weighing less than 15 kg.

Traditional ivermectin tablet formulations are not suitable for children weighing less than 15 kg, necessitating innovative approaches to ensure acceptability, safety and efficacy. There is a need for new, child-friendly formulations of ivermectin that can ensure accurate dosing, improved acceptability, palatability, safety during administration, and stability suitable for diverse environmental conditions where these NTDs occur. Young children are an extremely important population suffering a disproportionate health burden from helminth and scabies infection [6]. Previous studies have highlighted significant inter-individual variability in drug exposure with STROMECTOL^®^ [9,10], prompting a detailed comparison to ensure that new formulations can provide consistent and reliable therapeutic outcomes. Indeed, children weighing less than 15 kg that were treated with ivermectin doses less than 200 µg/kg were less likely to achieve therapeutic success for scabies compared to children treated with doses 200 µg/kg and above [11]. To address these needs, a novel orodispersible tablet (ODT) formulation of ivermectin, called CHILD-IVITAB, has been developed utilizing multifunctional template inverted particle (TIP) technology, designed to provide rapid disintegration, controlled absorption, and enhanced taste masking [12,13,14,15]. The rapid disintegration time (less than 10 s) of CHILD-IVITAB greatly improves ease of administration, virtually eliminates any choking risk, and removes need for potable water, which would facilitate use of ivermectin in children under 15 kg in MDA programs. To inform the pediatric program of CHILD-IVITAB, 16 healthy adults were enrolled in a phase I, single-center, open-label, randomized, two-period, cross-over, single-dose trial which aimed to compare the palatability, tolerability, and bioavailability and pharmacokinetics (PK) of CHILD-IVITAB compared against the marketed ivermectin tablets (STROMECTOL^®^) at a single dose of 12 mg in a fasting state. Non-compartmental analysis (NCA) demonstrated that CHILD-IVITAB yielded controlled absorption associated with reduced variability in drug exposure as compared to STROMECTOL^®^ [16]. The objective of the present study is to develop a population PK model to characterize the absorption profile and variability of the ivermectin CHILD-IVITAB formulation compared to the reference formulation STROMECTOL^®^. In addition, the developed model will be applied to simulate and evaluate dosing for a planned pediatric study in children ≥ 15 kg and those < 15 kg. By addressing these aspects, the goal is to contribute to optimizing ivermectin dosing strategies for children, ensuring consistent exposure and effective treatment outcomes across different age groups [7].

## 2. Materials and Methods

### 2.1. Study Design

The data originate from a previously published phase I, single-center, open-label, randomized, two-period, cross-over, single-dose trial (NCT05477810) [16], which aimed to compare the palatability, tolerability, bioavailability and pharmacokinetics of ivermectin ODT (CHILD-IVITAB) against the marketed ivermectin tablets (STROMECTOL^®^) at a single dose of 12 mg in a fasting state [10]. Subjects were instructed to place four ODTs of CHILD-IVITAB (3 mg) between the gum and the cheek for 30 s, then rinse and swallow with 150 mL of water. Subjects were instructed to immediately swallow four tablets of STROMECTOL^®^ (3 mg) with 150 mL of water. Both periods were separated by a wash-out period of at least 7 days. Palatability, tolerability, safety, pharmacokinetics and their variability were assessed in 16 healthy adult subjects. Power estimation can be found in the study by Dao et al. [16]. During each period, venous blood samples were collected in EDTA tubes pre-dose, and at 0.5, 1, 2, 3, 4, 5, 6, 8, 10, 24, 48, 72 and 96 h post-dose. Blood was centrifuged at 10 °C and 3220 g for 30 min and plasma was separated and stored at −20 °C until bioanalytical analysis. Plasma ivermectin samples were quantified by liquid chromatography tandem mass spectrometry (LC-MS/MS) [17]. The lower limit of quantification (LLOQ) and upper limit of quantification (ULOQ) of ivermectin plasma samples were 0.5 ng/mL and 250 ng/mL, respectively.

### 2.2. Pharmacometric Population PK Modeling

All participants who received the study drug during the study were included in the pharmacometric PK analysis of ivermectin, which used nonlinear mixed-effects modeling within the pharmacometric PK software Monolix (version 2023, Lixoft SAS, a Simulations Plus company, Antony, France). Model simulations were performed with the pharmacometric PK software Simulx (version 2023, Lixoft SAS, a Simulations Plus company). R version 4.3.1 was used within RStudio (version 2023.06.1, Vienna, Austria) for data handling, graphical visualization, and numerical calculations. Ivermectin plasma concentrations below the LLOQ and above the ULOQ were censored, except for pre-dose samples which were set to 0. Here, the CENSORING column in Monolix was used, corresponding to the M3 method in NONMEM [18].

### 2.3. Base Pharmacokinetic (PK) Model

For model building purposes, the population PK model developed by Brussee et al., using PK data obtained from 200 children 2–12 years of age and 11 adults, was used as the baseline structural model [7]. Brussee et al.’s model is a two-compartment PK model including two transit compartments to account for a delay in absorption, which was initially assumed for both formulations [19]. To align with the previously reported model from Brussee et al., body weight was included as a covariate for clearance *CL*, intercompartmental clearance *Q*, and volume of distribution in the central *Vc* and peripheral *Vp* compartment, and allometric scaling centered to 18 kg was applied, with the coefficients fixed to 0.75 for clearance and 1 for volume (Figure 1). Ivermectin typical oral clearance (*CL*) was described as a nonlinear function of weight (*CL* = 5.8 × (weight/18)^0.75^). The model included two transit compartments with *k_tr_* = *k_a_*, leading to a computed mean transit time *MTT* = 3/*k_tr_*. The choice of two transit compartments as the baseline model structure has been made because of a previously published model-based analysis in pediatric patients. However, another study in healthy volunteers has found an increased number of transit compartments to describe ivermectin absorption data best (N = 6) [9,10]. A sensitivity analysis for the number of transit compartments for each formulation, with up to 6 transit compartments tested, was conducted. Population parameters were estimated, using estimates from Brussee et al. as initial values. Again, as per Brussee et al., inter-individual variability (IIV) was included for the absorption rate constant, clearance, and both volume parameters (central and peripheral compartment), with individual parameters assumed to be log-normally distributed. Initially, a mixed residual error model was assumed. The covariance matrix of random effects was initially set to a diagonal matrix (no correlation between random effects assumed).

### 2.4. Alternative Investigated Model Structures

As structural models, one- and three-compartment models with first-order elimination were also evaluated. Additive, proportional and combined (i.e., additive and proportional) residual error models were investigated to describe the residual variability.

### 2.5. Correlation between Parameters

After selecting the structural model, the covariance matrix of random effects was built, starting from a diagonal matrix and then progressively assessing the significance of correlation terms by assessing scatterplots of the random effects and Pearson correlation coefficients. Screening for correlations was performed using “conditional distribution mode” in Monolix.

### 2.6. Investigation of a Formulation Effect and Potential Other Covariates

A potential formulation effect (CHILD-IVITAB or STROMECTOL^®^) on absorption rate *k_a_* (and resulting *MTT*) and/or relative bioavailability *Frel* (with *Frel* set to 1 for the reference formulation STROMECTOL^®^ and estimated *Frel* for CHILD-IVITAB) was evaluated with formulation-specific IIV. As the age range in Dao et al.’s trial was narrow (i.e., healthy young adults), this covariate was not investigated. Additionally, a gender covariate was investigated on the model parameter *CL*.

### 2.7. Evaluation of Population PK Model

A sensitivity analysis was conducted to evaluate the impact of specific individuals on model fit. Goodness of fit was graphically evaluated using standard plots (prediction vs. observations, randomness of residual scatter plots versus time/predictions and of random effects versus covariates) and a simulation-based visual predictive check of key models. Model development considered reductions in the objective function between candidate models (ΔOFV = −2 × log-likelihood), reductions in the inter-subject variability and residual error, and parameter precision and the clinical relevance of estimated effects. A visual predictive check (VPC) was performed, with n = 1000 simulations to evaluate whether the model can accurately predict the observed concentrations and capture the observed variability. A convergence assessment was conducted to assess the reproducibility of the results.

### 2.8. Model-Based Simulations to Evaluate Dosing of CHILD-IVITAB in Persons ≥ 15 kg and Children < 15 kg

Simulations were conducted to compare simulated ivermectin exposure *AUC*_0–96h_ (i.e., up to the last measured PK timepoint) and *AUC*_0–168h_ (i.e., extrapolated *AUC* up to 7 days after first dosing) following a single administration of 200 µg/kg of the reference formulation STROMECTOL^®^ in adults, and simulated ivermectin exposure (*AUC*_0–96h_ and *AUC*_0–168h_) in persons ≥ 15 kg and children < 15 kg following a single administration of various doses of CHILD-IVITAB. Different dosing scenarios were simulated with a dose of (i) 200 µg/kg; (ii) 250 µg/kg and (iii) 300 µg/kg. Monte Carlo simulations were performed using the developed PK model and population parameters to generate 1000 concentration–time profiles of ivermectin for each scenario, after which *AUC*_0–96h_ and *AUC*_0–168h_ were calculated by integration of simulated observed concentrations. Simulations of reference exposure (adults, STROMECTOL^®^ formulation, 200 µg/kg) were conducted for a population with a weight distribution (normal distribution) of N (mean, µ = 85.1 kg, standard deviation, σ = 11.7 kg), to yield a weight range of approximately 50.1 to 120.0 kg (comprising ± three standard deviations). Simulations of expected CHILD-IVITAB exposure in children under varying doses were similarly conducted in each weight group as follows: (i) for 5.0–7.5 kg, weight ~ N (µ = 6.3 kg, σ = 0.4 kg); (ii) 7.6-10.0 kg: weight ~ N (µ = 8.8 kg, σ = 0.4 kg); (iii) 10.1–14.9 kg: weight ~ N (µ = 12.5 kg, σ = 0.8 kg); (iv) 15.0-30.0 kg: weight ~ N (µ = 22.5 kg, σ = 2.5 kg); and (v) 30.1–50.0 kg: weight ~ N (µ = 40.1 kg, σ = 3.3 kg). As CHILD-IVITAB is dosed in 1 mg and 3 mg tablets, we established matching between fixed CHILD-IVITAB dosing and weight-based dosing, stratified by weight group.

## 3. Results

All sixteen healthy volunteers were included in the data analysis. Baseline demographics are shown in Table 1.

A total of 448 ivermectin venous plasma concentrations were available and included in the population PK analysis, of which 40 (8.9%) concentrations were below the limit of quantification (BLQ), all during the absorption phase, and 16 were pre-dose samples. The remaining plasma concentrations of ivermectin following CHILD-IVITAB and STROMECTOL^®^ dosing were in the range of 0.53–92.45 ng/mL and 0.51–82.79 ng/mL, respectfully (Figure 2).

### 3.2. Pharmacometric Population PK Modeling

Similar to Brussee et al. [7], a two-compartment PK model including two transit compartments to account for a delay in absorption, first-order elimination kinetics, weight-dependent clearance and distribution (allometric scaling centered to 18 kg), and a combined error model described the ivermectin data well for both CHILD-IVITAB and STROMECTOL^®^ formulations (Figure 1, Appendix A). Estimated PK parameters of the fitted two-compartment model are shown in Table 2. There are notable differences in the study design and patient demographics between the data included in this study and the data used to develop the referenced model. Brussee et al.’s analysis included extensive PK data from 200 children aged 2–12 years and 11 adults. Despite these differences in the demographics, the estimation of ivermectin clearance, the peripheral volume of distribution, and the absorption rate constant and transit rate constant were similar. De novo parameter estimation of central volume of distribution was ~52% lower, associated with ~67% higher intercompartmental clearance than in the model from Brussee et al. [7].

Decreasing variability in absorption parameters by approximately 50% was observed with CHILD-IVITAB (19%, RSE = 22%) compared to STROMECTOL^®^ (43%, RSE = 19%). Inter-individual variability in *CL, Vc, Vp*, and *Frel* of CHILD-IVITAB were estimated at 67%, 84%, 57%, and 61%, respectively (Table 2). Correlations between parameters with variability were suggested (*p*-value < 0.05) and set between *CL, Vc, Vp* and *Frel* in the correlation matrix, with estimated correlation values of 0.95 between *Vc* and *CL*, 0.89 between *Vp* and *CL*, 0.8 between *Frel* and *CL*, 0.92 between *Vc* and *Vp*, 0.84 between *Frel* and Vc, and 0.92 between *Frel* and *Vp*. Incorporating correlations between random effects improved the VPC (Figure 3).

### 3.3. Alternative Investigated Model Structures

A one-compartment model did not appropriately capture the elimination phase of ivermectin for both formulations. A three-compartment model did not yield an analytical solution for most PK parameters. A reduced model without transit compartments did not appropriately capture the maximal concentration *C_max_* for both ivermectin formulations. As expected, estimating distinct *k_tr_* and *k_a_* yielded model instability (RSE > 100%) due to over-parametrization. No IIV on *Frel* resulted in large RSE on random effects of *CL*, *k_a_* and *Vp* (>100%).

### 3.4. Investigation of a Formulation Effect and Potential Other Covariates

Ivermectin absorption was faster with CHILD-IVITAB than with STROMECTOL^®^ with typical *k_a_* values of 2.4 (95% CI: 2.17–2.68) and 1.6 (95% CI: 1.27–1.92) per hour, respectively, leading to a mean transit time (*MTT*) of 1.2 (95% CI: 1.12–1.38) hours with CHILD-IVITAB and 1.9 (95% CI: 1.56–2.36) hours with STROMECTOL^®^ (Table 2). The absence of allometric scaling in the model worsened the model fit (*p*-value of likelihood ratio test, LRT > 0.05 and residual standard errors (RSEs) of the random effects on *CL, Vc, Vp* and *Frel* > 100%). There was no formulation effect on apparent clearance (*CL*), with estimated *CL* of 5.2 (95% CI: 4.00–6.83) L/h and 5.9 (95% CI: 3.94–8.94) L/h for CHILD-IVITAB and STROMECTOL^®^, respectively, and therefore, this was not retained in the final population PK model. The effect of formulation on relative ivermectin bioavailability was not significant, with *Frel* of CHILD-IVITAB (1.30%, 95% CI: 0.97–1.74) not significantly differing from STROMECTOL^®^ (F fixed to 1, for no information from intravenous administration) (Table 2). Gender effect on *CL* was not a significant covariate to be included in the model (RSE > 100%).

### 3.5. Evaluation of Population PK Model

No subject was excluded from the main analysis. Individual clearance values ranged from IQR: 10.60 to 20.85 L/h. Individual #14 was excluded from the model sensitivity analysis, as particularly high oral clearance (65.83 L/h) and high (approximately 4-fold increased) relative bioavailability were estimated. Exclusion did not impact population parameter estimates significantly (overlapping 95% CI, Appendix A), but resulted in reduced observed IIV and a slightly better description of concentrations measured. The goodness-of-fit plots (individual and population predicted vs. observed concentrations, individual fits, individual parameter distribution plots and conditionally weighted residuals vs. population predications and time after dose) showed that the model in the sensitivity analysis described the data well (Appendix A). A sensitivity analysis for the number of transit compartments, with up to six transit compartments tested, showed that four transit compartments yielded the best model fit (lowest OFV). However, there was no change in estimated clearance compared to the base PK model with two transit compartments, and hence no effect on simulated *AUCs*. Adding a distinct number of transit compartments per formulation did not improve the model fit (Appendix A).

### 3.6. Model-Based Simulations to Evaluate Dosing of CHILD-IVITAB in Persons ≥ 15 kg and Children < 15 kg

From model-based simulations a reference exposure *AUC_0–96h_* and *AUC*_0–168h_ following a single administration of 200 µg/kg of the reference formulation STROMECTOL^®^ in adults was calculated to a median of 800 (inter-quartile range, IQR: 516–1230) µg·h/L and 884 (IQR: 572–1380) µg·h/L, respectively (Table 3).

A single dose of 200 µg/kg ivermectin CHILD–IVITAB was predicted to lead to 24.8%, 17.8% and 12.1% lower exposure (*AUC_0–96h_*) in children weighing 5.0–7.5 kg, 7.6–10.0 kg and 10.1–14.9 kg compared with adults receiving STROMECTOL^®^, respectively (Appendix A). Therefore, a dose adjustment in children < 15 kg was deemed necessary to achieve equivalent exposure coverage as in children ≥ 15 kg and adults. Model-based simulations indicated that a single dose administration of 250 µg/kg of CHILD-IVITAB is associated with equivalent exposure coverage in children < 15 kg compared with children ≥ 15 kg and adults receiving 200 µg/kg STROMECTOL^®^ (Appendix A). Evaluating the 300 µg/kg dosing regimen with CHILD-IVITAB yielded 12.9%, 23.4% and 31.3% over-exposure in children weighing 5.0–7.5 kg, 7.6–10.0 kg and 10.1–14.9 kg compared with adults receiving the reference 200 µg/kg of STROMECTOL^®^, respectively (Appendix A). Therefore, Table 3 and Figure 4 show ivermectin target exposure values obtained with the recommended dosing regimen using CHILD-IVITAB.

From Table 4, matching was established between the CHILD-IVITAB fixed-dosing regimen (given as 1 mg and 3 mg tablets) and weight-based dosing, stratified by weight group.

## 4. Discussion

Currently, there is a lack of a child-friendly, age-appropriate ivermectin formulation for young children, and ivermectin is currently not approved for children and infants < 15 kg [6,20,21]. Current oral administrations of ivermectin (crushed tablets in water or locally produced suspensions) for young children are prone to imprecise dosing due to loss of product after crushing and sedimentation of product after suspension. Furthermore, these methods are cumbersome and thus not appropriate for administration at scale during MDAs [22,23]. These factors may result in reduced drug adherence and effectiveness of ivermectin-based treatments in pediatric patient populations. As such, a child-friendly oral ivermectin formulation is needed. In a recent study in healthy adults [16], it was shown that palatability with CHILD-IVITAB was enhanced as compared to STROMECTOL^®^ which would improve acceptability during MDA. CHILD-IVITAB was well tolerated, and there were no adverse events reported in the study. Further, CHILD-IVITAB was associated with considerably reduced inter-individual variability in overall exposure (*AUC*_0–96h_ and *AUC*_0-inf_) with close to equivalent exposure coverage as compared to STROMECTOL^®^.

In this study, a previously published population PK model of oral ivermectin was used to characterize the absorption profile of the two formulations CHILD-IVITAB and STROMECTOL^®^ in healthy adults [7]. All PK parameter estimates were comparable with previously published results in children aged 2–12 years old [7]. Absorption with CHILD-IVITAB was faster than with STROMECTOL^®^ with typical k_a_ (set equal to k_tr_) values of 2.4 (95%CI: 2.17–2.68) vs. 1.6 (95% CI: 1.27–1.92) per hour, respectively, and corresponding shorter calculated mean transit time (MTT) with values of 1.2 (95% CI: 1.12–1.38) vs. 1.9 (95% CI: 1.56–2.36) hours, respectively. Some differences in model parameter estimates were observed when compared to values previously reported in children [7]. In particular, a 52% lower central volume of distribution and ~67% higher intercompartmental clearance were estimated, resulting in faster (×5) initial decline (alpha: 0.56 h^−1^ in the present study and 0.11 h^−1^ in Brussee et al. [7]), but a similar terminal elimination phase (beta: 0.028 h^−1^ in the present study and 0.023 h^−1^ in Brussee et al. [7]) (Appendix A). We hypothesize that these differences in the initial exposure profile could mainly be the result of different study designs, including different formulations used (ELEA ivermectin 3 mg or mini-tablets 0.5 mg produced at the Hospital Pharmacy of Basel University), administration in a fed versus fasted state in the present study, a different blood sampling approach (capillary versus venous blood sampling in the present study) and timing (more dense early sampling in the present study, and an additional late measurement at 96 h) [8]. A decreasing variability of absorption parameters (k_a_) by ~50% was seen with CHILD-IVITAB compared to STROMECTOL^®^ with the population PK model, in line with previous findings from non-compartmental PK analysis [16]. Age-dependent variation in the PK of ivermectin has been reported in previous studies. Decreased intestinal motility in children is thought to cause limited transit time (3–7.5 h in children vs. 3–4 h in adults), which in turn might explain the increased relative bioavailability [8]. In the near future, we plan to collect PK data in children to further characterize absorption in pediatric patients. From the planned pediatric PK study, we may be able to investigate potential age-dependent effects of ivermectin absorption. The marginally ~20% increased relative bioavailability estimated, as well as faster absorption process, could represent a small fraction of ivermectin absorbed buccally. However, we did not consider model-based analysis of such a possible parallel buccal absorption process, which may be defined in terms of the fraction of dose absorbed buccally with the corresponding buccal absorption rate constant. In fact, on an individual level, lower relative bioavailability was also estimated for CHILD-IVITAB for 7/16 subjects (Appendix A), which would imply an unphysiological negative fraction absorbed buccally. The faster absorption observed for CHILD-IVITAB (in the fasted state) could also be related to faster dissolution and gastric emptying, especially since absorption for ivermectin is assumed to be limited by solubility rather than permeability (classified as BCS class II). In the fed state, increased bile micelle-mediated solubility appears to explain faster and more complete absorption [24]. Further preclinical and clinical studies are being designed to focus on characterizing trans-buccal absorption process of the ODT. There was no formulation effect on apparent clearance (*p*-value > 0.05). A trend towards higher (~30%) ivermectin relative bioavailability with CHILD-IVITAB compared to STROMECTOL^®^ was observed, also confirming results from the trial [16]. One subject (#14) showed potentially low absolute bioavailability resulting in high oral clearance. Sensitivity analysis excluding subject #14 showed a significant decrease in relative bioavailability. Unknown and potentially variable absolute oral bioavailability may also explain the high correlation between estimated individual PK parameters. Inspection of individual profile plots (Figure 1) suggested that IIV is necessary to be included in models due to mostly higher CHILD-IVITAB profiles, but in a few instances, also lower profiles compared to STROMECTOL^®^. As such, variability in ivermectin exposure between these two formulations does not seem to originate from differences in drug clearance but is driven by a more controlled absorption process with CHILD-IVITAB. This would result in more homogeneous response to ivermectin, limiting under- and over-exposed patients, and may consequently be beneficial in a clinic or in MDA settings.

### 4.1. Modeling and Simulation to Facilitate Dose Selection for a Pediatric Study in Children with Scabies

Clinical trials are being set up in LMICs (EPIC-15 trial in Brazil, NCT06404333) and Europe to assess palatability, tolerability, safety, efficacy, and exposure coverage of CHILD-IVITAB in pediatric patients with a parasitic disease, including children weighing more than 5 and less than 15 kg. The objectives of this clinical study were to determine drug exposure coverage after oral administration of CHILD-IVITAB applying weight-adjusted dosing in children > 5 kg and <15 kg compared to children ≥ 15 kg treated for scabies. The popPK model developed by Brussee et al. for standard ivermectin in children 2–12 years of age indicated that an increased dose of 250 and 300 µg/kg would be needed in school-aged children (6–12 years) and pre-school-aged children (2–5 years), respectively, to achieve equivalent exposure coverage in children compared to adults [7]. The current developed popPK model was used to perform simulations to support the design of such a clinical trial, with the objective to determine ivermectin concentration profiles and drug exposure coverage for 200 µg/kg, 250 µg/kg and 300 µg/kg, in children ≥ 15 kg and <15 kg following a single administration of CHILD-IVITAB. Outputs from simulations in children < 15 kg revealed that a dose of 250 µg/kg CHILD-IVITAB is associated with consistent exposure coverage as compared to a dose of 200 µg/kg in children ≥ 15 kg and adults, up to 7 days post administration. CHILD-IVITAB will be developed at 1 mg and 3 mg strengths, allowing for fine-tuned dosing depending on body weight in children < 15 kg by combining both fixed-dose regimens of 1 mg and 3 mg [25]. Altogether, these results suggest that CHILD-IVITAB is a suitable formulation for clinical or MDA settings to prevent and/or treat NTDs (e.g., scabies, helminth, onchocerciasis), not just in adults, but also in adolescents and children ≥ 15 kg and children and infants < 15 kg.

### 4.2. Limitations

One of the limitations of this study is the relatively small sample size. Nevertheless, the rich sampling design enabled parameter estimation with good precision. Further pediatric studies will be conducted to collect safety data in children. In this study, we considered weight (allometric scaling) to predict exposure in pediatric patients up to the age of 6 months (i.e., approximately 5 kg), while age might generally play a role in the metabolization pathway due to maturation processes in the smallest age group. However, cytochrome P450 3A4 (CYP3A4) is the predominant isoform responsible for the metabolism of ivermectin by human liver microsomes, and after 2 years of age, CYP3A4 exhibits 100% of its activity [26]. In addition, no particular effect of age on P-glycoprotein (P-gp) is expected [27]. Therefore, while age might be an important covariate to consider below 2 years, age will likely have a limited impact on ivermectin elimination in the considered target population (from 2 years of age, i.e., approximately 10 kg), as found previously [7]. Further clinical studies are warranted to expand knowledge on ivermectin exposure in pediatric patients < 10 kg. While VPC showed overall adequate model fit, some trend for underprediction of late concentrations measured at 96h may need to be acknowledged (residual plots, Appendix A). The presented model should therefore be used with caution for extrapolation to later timepoints (e.g., 7 days after administration). The trend of underprediction of late concentrations may be reduced by fitting a three-compartmental model; however, we could not estimate corresponding parameters reliably (RSE > 100%, OFV = 2704 vs. 2341 for base population PK model, CV > 100% on parameter estimates). Further clinical studies may help to identify such a model structure, with planned concentration measurements after 96 h. Elimination of the need for water administration is one of the main advantages of CHILD-IVITAB. However, in the settings of the presented trial in healthy adults, CHILD-IVITAB was administered for 30 s in the buccal cavity, then rinsed and swallowed with water. In the case of CHILD-IVITAB, the drug may partly be absorbed trans-buccally within the first 30 s after uptake, before water intake. Further preclinical and clinical studies are being designed to focus on characterizing the sublingual, trans-buccal and gastro-intestinal absorption process of the ODT. Nevertheless, the faster absorption process, as seen with the novel template inverted particle (TIP) technology loaded with ivermectin CHILD-IVITAB, shows significant potential for other APIs that require a fast onset of action.

## 5. Conclusions

This pharmacometric analysis confirmed that CHILD-IVITAB shows faster and more controlled absorption than STROMECTOL^®^ in the fasting state, explaining previously reported reduced variability in ivermectin exposure with CHILD-IVITAB. Model-based simulations indicated that a CHILD-IVITAB dose of 250 µg/kg in children < 15 kg is expected to achieve equivalent ivermectin exposure coverage in these children as compared to children ≥ 15 kg and adults treated with a dose of 200 µg/kg.

## Figures and Tables

**Figure 1 pharmaceutics-16-01186-f001:**
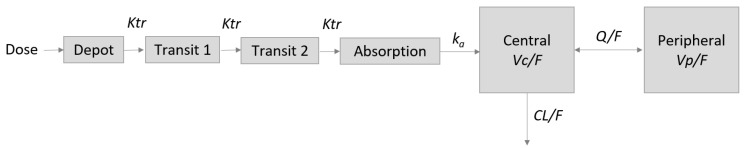
Graphical representation of the two-compartment PK model best describing the ivermectin data for both CHILD-IVITAB and STROMECTOL^®^ formulations. The model included weight-dependent clearance and distribution (allometric scaling centered to 18 kg) and two transit compartments. *CL*/*F*: apparent clearance, *F*: bioavailability, *k_a_*: absorption rate constant, *k_tr_*: transfer rate constant (*k_tr_* = *k_a_*), *MTT*: mean transit time (*MTT* = 3/*k_tr_*), *Q*/*F*: apparent intercompartmental clearance, *Vc*/*F*: apparent volume of the central compartment, *Vp*/*F*: apparent volume of the peripheral compartment.

**Figure 2 pharmaceutics-16-01186-f002:**
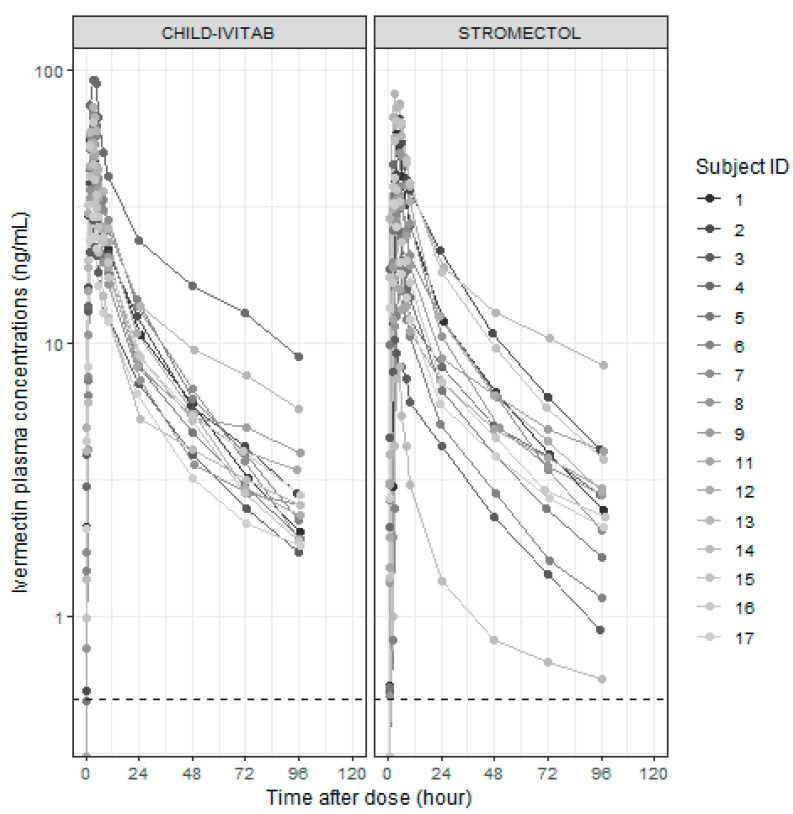
Individual observed plasma concentrations versus time after single oral dose of 12 mg ivermectin. Dotted horizontal line corresponds to LLOQ = 0.5 ng/mL. Note: one screening failure occurred (individual #10) because of positive cannabis drug screen.

**Figure 3 pharmaceutics-16-01186-f003:**
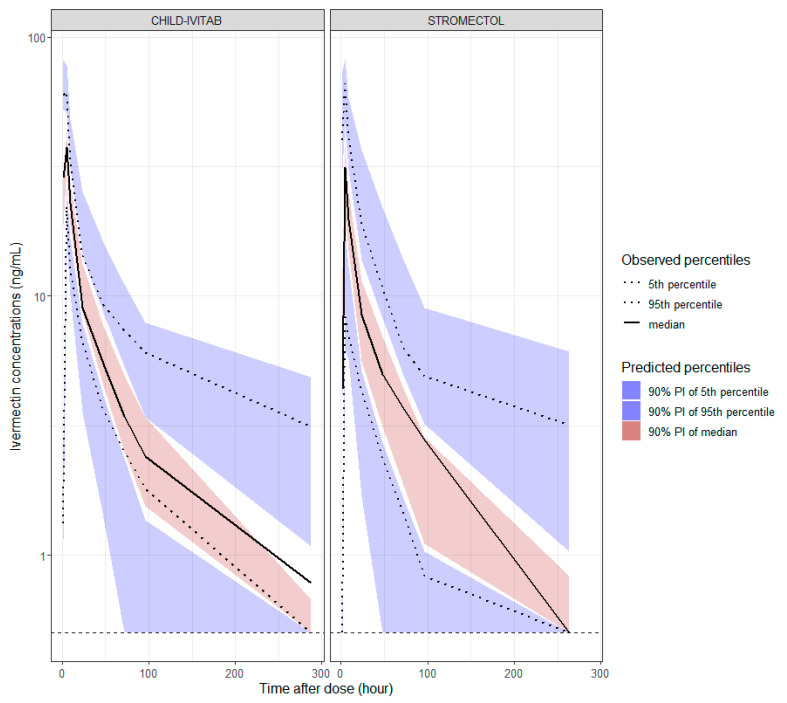
Visual predictive checks for PK model presented in Table 2 for ivermectin concentrations (ng/mL) on *y*-axis (log scale) and time after dose (h) on *x*-axis. Dotted horizontal line corresponds to LLOQ = 0.5 ng/mL. Pre-dose samples of next dosing from cross-over trial were included in the VPC.

**Figure 4 pharmaceutics-16-01186-f004:**
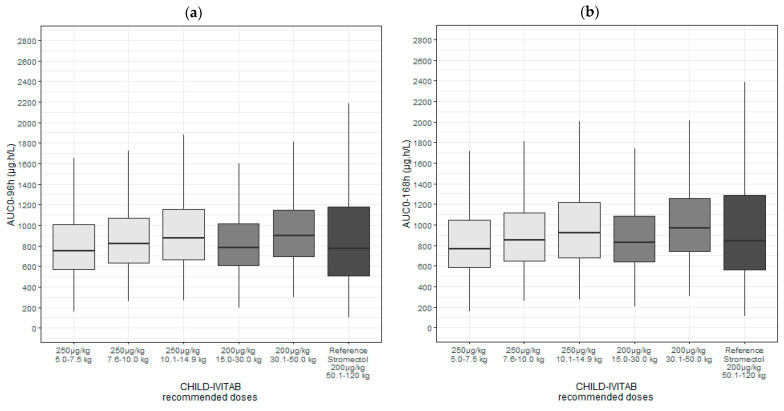
Simulated ivermectin exposure (N = 1000) according to weight-based ivermectin dosing regimen in adults following a single administration of 200 µg/kg of the reference STROMECTOL^®^ formulation and according to defined dose recommendations of 250 µg/kg in children < 15 kg and 200 µg/kg in children ≥ 15 kg following a single CHILD-IVITAB administration. (**a**) Simulated ivermectin exposure 96h after dosing (area under the concentration–time curve, *AUC*_0–96h_). (**b**) Simulated ivermectin exposure 168h after dosing (*AUC*_0–168h_). IQR, inter-quartile range.

**Table 1 pharmaceutics-16-01186-t001:** Demographic characteristics of study participants included in population PK analysis. Represented as median [IQR, inter-quartile range] or n (%). BMI, body mass index.

Demographic Characteristics of Study Participants (N = 16)	
Age (years)	24.0 [20.8, 28.0]
Weight (kg)	63.7 [58.0, 71.5]
Height (cm)	171 [168, 178]
BMI (kg/m^2^)	22.3 [19.7, 23.1]
Gender	
Female	7 (43.8%)
Male	9 (56.3%)
Ethnicity	
African	2 (12.5%)
Caucasian	10 (62.5%)
Hispanic/Latin American	1 (6.3%)
Multiracial	3 (18.8%)

**Table 2 pharmaceutics-16-01186-t002:** Population PK parameter estimates for ivermectin. Proportional and additive errors are reported as variance estimates (σ^2^). *CL*, clearance; *Frel*, relative bioavailability (*Frel* set to 1 for the reference formulation STROMECTOL^®^ and estimated *Frel* for CHILD-IVITAB); IIV, inter-individual variability, reported as coefficient of variation (CV%); *Q*, intercompartmental clearance; *k_a_*, absorption rate constant; *k_tr_*, transfer rate constant; RSE, relative standard error; *Vc*, volume of distribution in the central compartment; *Vp*, volume of distribution in the peripheral compartment. Additive error (mg/L): 0.69 (15%); proportional error: 0.16 (7%).

Parameter (Unit)		Value (RSE %) (Shrinkage)
CHILD-IVITAB absorption rate constant and transit rate constant (h^−1^)	*k_a_ = k_tr_*	2.41 (6%)
STROMECTOL^®^ absorption rate constant and transit rate constant (h^−1^)		1.56 (12%)
CHILD-IVITAB mean transit time (h)	*MTT =* 3/*k_tr_*	1.24
STROMECTOL^®^ mean transit time (h)		1.92
Clearance (L/h)	*CL*	5.8 (17%) × (WT/18)^0.75^
Volume of distribution (L)	*V_c_*	60.29 (24%) × (WT/18)
	*V_p_*	103.56 (16%) × (WT/18)
Intercompartmental clearance (L/h)	*Q*	9.73 (18%) × (WT/18)^0.75^
CHILD-IVITAB relative bioavailability	*Frel*	1.30 (16%)
IIV k_a CHILD-IVITAB_ (CV%)		0.19 (22%) (14%)
IIV k_a STROMECTOL_ (CV%)		0.43 (19%) (0.70%)
IIV CL (CV%)		0.67 (27%) (1.6%)
IIV V_c_ (CV%)		0.84 (28%) (1.1%)
IIV V_p_ (CV%)		0.57 (41%) (5.9%)
IIV *Frel* (CV%)		0.61 (24%) (2.3%)

**Table 3 pharmaceutics-16-01186-t003:** Simulated ivermectin exposures (area under the concentration–time curve, *AUC_0–96h_* and *AUC*_0–168h_) according to weight-based ivermectin dosing regimen. Simulated ivermectin exposures in 1000 adults following a single administration of 200 µg/kg of the reference STROMECTOL^®^ formulation and according to defined dose recommendations of 250 µg/kg in children < 15 kg and 200 µg/kg in children ≥ 15 kg following a single CHILD-IVITAB administration. IQR, inter-quartile range.

		Weight-Based Ivermectin Dosing Regimen		
Ivermectin Formulation	Body Weight (kg)	Recommended Dosing Regimen (µg/kg)	Simulated Median *AUC*_0–96h_ [IQR] (µg·h/L)	Simulated Median *AUC*_0–168h_ [IQR] (µg·h/L)
**STROMECTOL** ^®^ **(reference in adults)**	50.1–120	200	800 [516, 1230]	884 [572, 1380]
**CHILD-IVITAB**	5.0–7.5	250	752 [574, 1010]	770 [589, 1040]
	7.6–10.0	250	822 [635, 1070]	853 [651, 1120]
	10.1–14.9	250	879 [661, 1150]	919 [682, 1220]
	15.0–30.0	200	780 [612, 1010]	832 [639, 1080]
	30.1–50.0	200	899 [698, 1140]	971 [740, 1260]

**Table 4 pharmaceutics-16-01186-t004:** CHILD-IVITAB dose sliding scale stratified by weight group.

Body Weight (kg)	CHILD-IVITAB 1 mg Tablets	CHILD-IVITAB 3 mg Tablets	Effective Dose Range (µg/kg/dose)
5.0–7.5	1 to 2	-	133–400
7.6–10.0	2	-	200–263
10.1–14.9	-	1	201–297
15.0–30.0	-	2	200–400
30.1–50.0	-	3	180–300
50.1–120	-	4	100–240

## Data Availability

The data that support the findings of this study are available from the corresponding author upon reasonable request.

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
