# Peer review of "Pharmacometrics to Evaluate Dosing of the Patient-Friendly Ivermectin CHILD-IVITAB in Children ≥ 15 kg and <15 kg"

_pharmaceutics, 2024, doi:10.3390/pharmaceutics16091186_

Round 1

Reviewer 1 Report

Comments and Suggestions for Authors

The authors mention rate of absorption in the abstract to be 2.41 and 1.56. The rate of absorption is variable. This should be rate constant. 

Why are you using two transits in the mathematical model? Why would this transit apply to pediatrics? 

There is no discussion of sublingual buccal absorption and GI absorption since you have two different dosage forms.  

You excluded subject 14 from the study. What about subject 4?

How does age and weight affect your model? Can you share your mathematical model? 

Comments on the Quality of English Language

A minor review of English is warranted. 

Reviewer 2 Report

Comments and Suggestions for Authors

The manuscript “Pharmacometrics to evaluate dosing of the patient-friendly 2 ivermectin CHILD-IVITAB in children ≥ 15 kg and < 15 kg” proposes original modeling research with clinical relevance, e.g. it provides solid basis for the dose adjustment in the pediatric population for CHILD-IVITAB administration.

The proposed work shows sound application of population PK modeling. Below are suggestions which can further strengthen the manuscript:

  1. Quality of figures should be improved as currently it is hard to follow. In particular:
    1. Figure 2 – quality should be improved, I would suggest changing individual PK profiles for spaghetti plots, as it is hard to visualize population variability from individual data
    2. Currently there are no diagnostic plots in the main text. I would suggest adding at least VPC in the main text to provide more c readability for the final model.
    3. VPC plots quality is low, not possibly to assess model prediction quality, one suggestion to use log Y scale
    4. There is a potential model misfit, which could be seen at the residual analysis (Figure S1, B-C-D), authors should discuss this misfit to clarify that this doesn’t affect the prediction power of the model.  
    5. Individual fits, individual parameter distribution plots could be added in the supplementary
  2. Some important information about the model is missing. Could authors add/clarify the following issues:
    1. It could be good to show and discuss potential differences with the referenced model Brussee et al. and de novo parameter estimates.
    2. It could be good to test additional methods of BLOQ handling, e.g. M3, given the presence of such data points in absorption phase and low number of individuals that can affect parameter estimates.
    3. Shrinkage of the parameters are not shown, please provide estimates
    4. It is not clear which methodology was used for the estimation of non-diagonal elements, if conditional distribution implemented in Monolix was used this should be shown, as this can affect the estimates.
    5. I couldn’t figure out which residual error model was chosen as final (in table two additive and proportional parameters are shown), this should be better clarified
Comments on the Quality of English Language
    1. I would propose for Authors to change term pharmacometrics (PMX) model for population PK. There is only population PK model in the proposed study, so no need to use more generic term.

Reviewer 3 Report

Comments and Suggestions for Authors

Dear authors,

Thank you very much for providing this useful article that should add safety to application of drugs in pediatrics globally.

This article is nicely written and is scientifically based. Congratulations to the authors.

I have several major comments. The study population is very small (only 16 individuals) and consists of young adults with their representative PK parameters which may be different from that of small children. The data are compared with another trial from children. The questions I have are, did the small sample size allow to come up with solid results and conclusions.

Moreover, can the authors elaborate on the applicability of the PK parameters in children’s population.

There is also a large bias when comparing data from children who participated in another study.

Having these large limitations on board, how solid are the results and the conclusions from this article?

Wish you all the best in the revision process

Best regards

Reviewer

Round 2

Reviewer 3 Report

Comments and Suggestions for Authors

Dear authors, 

Thank you for addressing point-by-point suggestions and comments. 

I have no further significant remarks but some minor issues that I'm sure will also be captured in the editing process and publication (see below)

wish you all the best 

Regards

Reviewer

Page 3. Line 97.

Please provide a reference in this sentence “Power estimation can be found 97 in Dao et al.”

Page 4. Line 166.

Please modify the wording of “PMX population PK model” to “population PK model” in title as in the rest of the body of the text.

Page 15. Line 539.

Please check the accuracy of the reference N°3 and complete the necessary information

Author Response

Dear Reviewer,

We appreciate your positive feedback and have made the adjustments to the manuscript as suggested:

  • Page 3, line 97: we have added a reference to Dao et al. as requested.
  • Page 4, line 166: we have modified "PMX population PK model" to "population PK model" in the title to ensure consistency throughout the text.
  • Page 15, line 539: we have reviewed and corrected Reference #3 and added missing information.

We hope these changes meet your requirements and help to move the publication process forward.

Thank you once again for your consideration and guidance.

Best regards,

Klervi GOLHEN